# The Role of Gender Norms in Shaping Adolescent Girls' and Young Women's Experiences of Pregnancy and Abortion in Mozambique

Sally Griffin [1],*, Málica de Melo [1], Joelma Joaquim Picardo [1], Grace Sheehy [2], Emily Madsen [2], Jorge Matine [3] and Sally Dijkerman [2]

1  International Centre for Reproductive Health, Maputo 1100, Mozambique; m.demelo@icrhm.org.mz (M.d.M.); j.joaquim@icrhm.org.mz (J.J.P.)
2  Ipas, Chapel Hill, NC 27515, USA; sheehyg@ipas.org (G.S.); madsene@ipas.org (E.M.); dijkermans@ipas.org (S.D.)
3  Ipas, Maputo 1100, Mozambique; matinej@ipas.org
*  Correspondence: sallygriffin@proton.me; Tel.: +258-849-031-054

**Abstract:** Adolescents and young women in Mozambique experience high levels of unintended pregnancies, with induced abortion being a common outcome. Stigma and gender norms are likely to negatively impact experiences of pregnancy and abortion, and hamper access to information and services. We assessed knowledge, attitudes, practices, and experiences around pregnancy and abortion in six communities in Nampula and Zambézia provinces. We conducted 19 triad interviews with young women and girls, 19 focus group discussions with male and female adult community members, and 15 in-depth interviews with young women with abortion experience. Participants described how gender values, norms, and practices affect girls' risk of unintended pregnancy and their experiences of pregnancy and abortion. The drivers of adolescent pregnancy included transactional sex and gender-based violence, including early marriage, and gender roles and expectations that lead parents and others to oppose contraception. Stigma around abortion, early or unintended pregnancy, and adolescent sexuality is fueled by gender norms and contributes to girls seeking unsafe abortions. Pregnancy and abortion decision making often involves male partners and family members. In conclusion, gender norms strongly influence the occurrence and outcome of unintended pregnancies and abortion in Mozambique. While abortion legislation was recently liberalized, gender values, norms, and practices inhibit young women's and girls' access to services and need to be addressed in policy and programming.

**Keywords:** adolescent sexual and reproductive health; gender norms; stigma; adolescent pregnancy; induced abortion; Mozambique

## 1. Introduction

### 1.1. Role of Gender Norms in Shaping Sexual and Reproductive Health Outcomes

Gender inequality is an important social determinant of health and strongly influences sexual and reproductive health (SRH) outcomes [1,2]. Social norms, including gender norms, are a key driver of gender inequality and are a significant barrier to SRH choices and care, particularly when interacting with other determinants, such as age, disability, and poverty [3–5]. Gender norms are "the often unspoken rules that govern the attributes and behaviours that are valued and considered acceptable for men, women, and gender minorities" [6]. They are learned in childhood and maintained in the family and broader social context, through real or perceived social rewards and sanctions [7,8]. Gender norms are based on underlying values (such as ideologies of men's superiority), and result in many harmful practices, such as early marriage and gender-based violence (GBV) [8]. Examples of gender values and norms that can affect women's and girls' SRH decisions and

experiences include gendered cultural expectations around childbearing and motherhood, stigma around premarital sex and adolescent pregnancy, and beliefs and expectations around men's and women's sexuality [4,9,10]. Gender norms such as these have been shown to impact a wide range of SRH issues, including contraception, safe abortion, HIV, and GBV [2,9,10].

In the case of adolescents and young women, gender inequality interacts with inequalities and norms relating to age. Social and gender norms around sexual activity and pregnancy in adolescence work together to limit access to SRHR (sexual and reproductive health and rights) education and contribute to poor communication between adolescents and parents about sexuality and SRHR [11]. These norms also restrict adolescents' access to SRH services, including due to privacy concerns and provider bias, for example, denial of contraception for adolescents [12,13]. These gender and age barriers contribute to high levels of unintended pregnancy in sexually active adolescents and, consequently, a high demand for abortion in this age group [14].

Access to safe abortion care is an essential element of SRHR. Access to care is restricted by abortion stigma, "a shared understanding that abortion is morally wrong and/or socially unacceptable", which can be internalized, perceived, or enacted [15,16]. There is evidence that abortion stigma is partly driven by gender norms and is felt and enacted more strongly in the case of adolescents and young women in many settings [14,17].

### 1.2. Abortion in Mozambique

Mozambique is located on the east coast of Southern Africa, with an estimated population of over 33 million [18], of which 24% are adolescents [19]. The population is culturally diverse with over 40 languages spoken, and in terms of religion, 60% is Christian and 19% Muslim [19], the latter mainly in the coastal and northern areas. While the country is endowed with rich natural resources, most of the population relies on subsistence agriculture and fishing, particularly the majority that live in rural areas, and poverty levels are high [18]. In general, the northern provinces have poorer socioeconomic indicators than those in the south. Having won independence from Portugal in 1975, Mozambique entered a period of civil war until 1992 and, since then, has experienced pockets of conflict, most recently in Cabo Delgado province. Mozambique is also at high risk of environmental and climate-related crises. Both conflict and natural disasters can impact gender dynamics [20]. Harmful gender norms are prevalent in Mozambique and play a key role in driving the high levels of gender inequality that are seen in the country [21].

In Mozambique, over a quarter (26%) of adolescents aged 15–19 have an unmet need for contraception, which results in a high rate of adolescent pregnancy, many of which are unintended: 44% of 15- to 19-year-old girls become pregnant before turning 18, and 14% before they are 15 [22]. An estimated 24% of pregnancies in Southern Africa end in abortion, either safe or unsafe [23], and complications from unsafe abortions contribute to Mozambique's persistently high maternal mortality ratio, estimated at 289 deaths per 100,000 live births in 2017 [24]. One study found that postabortion clinical admissions represented more than 55% of all obstetric complications [25], while another estimated that abortion-related complications were responsible for between 11% and 18% of institutional maternal deaths in adolescents in the country [26]. Prior to 2014, abortion under any circumstance was criminalized, although "quasi-legal" services were provided in some larger hospitals, following a Ministry of Health (MoH) decree in 1981 [27]. Clandestine services were also provided for a fee in some health facilities, although many women and girls continued to seek alternatives outside of health facilities, and complications were frequent [28,29].

In 2014, recognizing the contribution of unsafe abortion to maternal mortality, Mozambique passed legislation that decriminalized abortion for women in the first 12 weeks of pregnancy, or 16 weeks in the case of rape or incest and 24 weeks in the case of fetal impairment (Law no. 34/2014). Following this legislative change, in 2017, the MoH published clinical and legal guidelines to support the provision of comprehensive abortion

care. The clinical guidelines cover safe abortion and postabortion care, closely following WHO guidelines. Safe abortion services can be offered only at designated health facilities by health workers trained in the clinical guidelines. Abortions must be performed with the consent of the pregnant woman, and minors must have consent from their parent, guardian, or another trusted adult. Since the publication of the clinical guidelines, safe abortion services have been slowly rolled out across the country.

Despite these important steps to improving access to safe abortion care in Mozambique, there is emerging evidence that significant barriers to access remain, including women's low knowledge of the legal situation and available services, their lack of autonomy to decide to terminate a pregnancy, and the poor availability of local abortion services [30]. While evidence from Mozambique since decriminalization is still limited, evidence from other countries indicates that it is likely that various other factors will continue to inhibit access to safe abortion care despite the legal change [31]. At the health facility level, key reported barriers include distance to services and insufficient equipment and supplies [32] and actual or perceived cost of formal services [33,34]. Provider-level barriers include provider bias, inadequate provider training, poor patient–provider relationships, limited human resource capacity, and high levels of conscientious objection in some settings [32,35–37]. There is often a lack of community knowledge and uncertainty about the legality of abortion and how and where to obtain care [34,36,38]. Social stigma together with religious and moral antiabortion views affects access to information and services [32,34], for example, creating a need for secrecy and privacy [36,39]. Most of these barriers are exacerbated for younger women, who experience heightened stigma and lack of knowledge, power, and resources [17].

### 1.3. Study Justification and Objectives

Given the global context, where, across several regions, there is a trend towards liberalization of abortion laws, it is fundamental to better understand the other factors—beyond the legal and policy context—that impact women's and girls' access to care. While the impact of gender norms on SRHR is generally recognized, few studies have examined in depth the gender-related factors that are likely restricting adolescents' access to safe abortion and other SRH services [40]. Furthermore, SRHR policy and programming for adolescents often does not take into account the impact of social and gender norms [3,4,41]. Understanding how gender norms affect young women's and adolescents' experiences of pregnancy and abortion is critical for developing programs and policies that increase their access to services and support. Mozambique provides a key opportunity to study these issues in a context where abortion has been partially decriminalized.

This study was carried out in the context of a project that aimed to increase access to comprehensive abortion care (CAC) and information in two provinces of Mozambique. To inform project interventions, the study aimed to assess adolescents', young women's, informal abortion providers', and other adult community members' knowledge, attitudes, practices, and preferences related to abortion in six communities in Nampula and Zambézia provinces. In this paper, we specifically explore findings on the gender values and norms that affect adolescents' and young women's experiences of pregnancy and abortion.

## 2. Materials and Methods

This was an exploratory qualitative study using triad interviews and individual in-depth interviews (IDIs) with adolescents and young women, focus group discussions (FGDs) with adult community members, and in-depth interviews with informal abortion providers. For this paper, we draw on data from the triad interviews and IDIs with adolescents and young women and the community FGDs.

### 2.1. Study Setting

The study team conducted this research in six communities in Zambézia and Nampula, neighboring provinces located in central and northern Mozambique, respectively. These

are the country's most populous provinces, with almost half of the national population. They rank poorly in most socioeconomic indicators compared with the national averages, including with regard to early marriage, contraceptive use, and girls' education [22,42]. Within each province, we selected sites to represent a range of urban and rural contexts. All sites were within the catchment area of health facilities where Ipas is supporting CAC provision, and were also selected based on the presence of the *Rapariga Biz* program, a partner of Ipas, which provides SRH education and services to girls. The study sites are described in Table 1 below.

**Table 1.** Study sites.

| Province | Communities | Site Characteristics |
|---|---|---|
| Nampula | Nampula City: Napipine, Gorongoza, Mutaunha, and 22 de Agosto neighborhoods | Urban and periurban neighborhoods on the outskirts of Nampula City |
| | Mogovolas District: Meluli B, Meluli C, and Namacarro B neighborhoods | Semirural and rural neighborhoods in and on the outskirts of Nametil town |
| | Nacala Porto District: Mucone and Ribaué neighborhoods | Semirural neighborhoods of Nacala Porto municipality |
| Zambézia | Quelimane City: Sangariveira neighborhood | Periurban neighborhood on the outskirts of Quelimane City |
| | Mocuba town: Samora Machel neighborhood | Semirural neighborhood on the outskirts of Mocuba town |
| | Mocuba District: Mugeba locality | Semirural and rural neighborhoods of Mugeba town |

*2.2. Study Participants and Procedures*

Data were collected during July and August 2018 by two female researchers and four young (under 25) female research assistants. The two researchers were each responsible for supervising the field work in one province and for supervising two research assistants; they also conducted some interviews in Portuguese. The research assistants were responsible for conducting interviews in local languages, with support from the more senior researchers. The data collection team members were trained by the principal investigator and two researchers on the use of interview guides, study procedures, ethical considerations, and conducting research with adolescents and on sensitive issues. The triad interview and FGD study instruments were tested prior to data collection in a separate location from the study sites. Facilitators used a semistructured interview guide to conduct discussions and interviews, while a second team member observed the interview and took notes. Interviews were conducted in Portuguese or in the locally spoken language (Macua in the Nampula sites and Chuabo in Zambézia). All interviews were audio-recorded and conducted in a private space to ensure visual and auditory privacy. Local *Rapariga Biz* supervisors and mentors assisted in the recruitment of study participants, helping to create an environment of trust in which young women felt safe and able to disclose sensitive personal information. For each type of interview, a sample size was determined that was estimated to be sufficient to achieve saturation, based on previous experience. Efforts were made during sampling to include diversity in terms of urban/rural residency, marital status, age, and other factors. During data collection, the team held daily reflection meetings to identify key findings and any issues to be addressed or adjustments necessary to the study tools or procedures.

*FGDs with adult community members.* The aim of the FGDs was to generate information and discussion about community practices, values, and norms in relation to abortion and community structures supporting or hindering abortion care. Participants were adult community members, including men and women of different ages and socioeconomic statuses within the community. Targeted and convenience sampling was used to identify participants, with the assistance of *Rapariga Biz* program staff. To be eligible for the FGDs, participants had to be 18 years old or over and a resident in the study area for at least 3 months. FGDs were conducted separately with men and women, and each group had between 7 and 15 participants. During the discussion, a narrative of a fictional girl, "Fatima", was used to encourage discussion regarding girls' experiences and the social norms surrounding girls who have an abortion. A participatory activity, community

resource mapping [43], was used to obtain and discuss information about where young people access reproductive health information and services, including abortion care, and why. Images of abortion pills were used to assist in identifying the drugs available locally.

*Triad interviews with adolescent girls and young women.* Triad interviews are in-depth interviews with three participants, which combine some of the benefits of individual IDIs with those of FGDs. In this case, "affinity triads" were interviewed, consisting of three friends or "intimate peers" [44]. The closeness of triad participants and their shared experiences and social context, together with evidence that triad participants are more likely to open up with close peers than with strangers in an FGD [44], make this an appropriate method to use for the discussion of sensitive issues, such as abortion. The aim of the triad interviews was to obtain information about young women's and girls' pregnancy and abortion knowledge, attitudes, decision-making pathways, practices, and preferences. To be eligible for the triad interviews, participants had to be female, aged between 15 and 24 years old, and a resident in the study area for at least 3 months. Participants were identified through targeted and convenience sampling, assisted by *Rapariga Biz* staff. For each triad, one young woman was recruited and asked to come to the interview with two friends, with the aim of creating an open and safe environment where they would feel comfortable sharing their experiences and thoughts. The participants were recruited to represent young women with different demographic characteristics (for example, in and out of school, married and unmarried, different age groups, and socioeconomic status). As in the FGDs, a fictional narrative and participatory mapping exercise were used to encourage discussion, and images were used to assist in identifying abortifacient drugs.

*Individual IDIs with young women.* The IDIs were carried out in a second phase, after the FGDs and triad interviews, with young women who had experienced an abortion. The objective was to understand the pathways for abortion care and the experiences of women who had abortions in and outside health facilities. The eligibility criteria were being female, between 15 and 24 years old, a resident in the community for at least 3 months, and having experienced an induced abortion within the previous 2 years (regardless of place or type of provider). Participants were identified through the triad interviews and FGDs using snowball sampling to identify potential participants who met the eligibility criteria. The subsequent contact with the potential participant was made by the person who had identified them, and in case they agreed to participate, the participant then contacted the research team or vice versa to schedule the interview.

### 2.3. Data Management and Analysis

Audio recordings were transcribed verbatim in Portuguese, with simultaneous translation from local languages where necessary. The study team checked the quality of the transcripts, which were subsequently translated into English. Two members of the ICRH-M research team coded and analyzed the Portuguese transcripts using the framework method for data analysis [45], with the assistance of a qualitative software program (NVivo 10, QSR International). First, we read a sample of the transcripts to obtain an overview of the data. Following this, an initial coding scheme was developed deductively, based on the literature and the research questions. We then read the transcripts carefully and indexed the data from each transcript. The coding scheme was iteratively adapted during the indexing process wherever new or contradictory themes emerged from the data. Following indexing, we identified themes emerging from the data, assigning meaning to emergent themes and concepts. For this paper, further to the initial analysis, we conducted specific analysis of the findings in relation to gender-related factors affecting experiences of pregnancy and abortion, using an adaptation of the RinGs gender analysis framework [46] to assess the following dimensions: access to and control over assets and resources; roles, responsibilities, and participation; cultural norms, beliefs, and practices; and power and decision making.

During data analysis, a subset of transcripts was coded by two researchers, and data analysis software was used to transparently document and share the coding process. The data analysis team met regularly to discuss adaptations to the coding framework,

and wider team discussions were held to interpret and draw conclusions from the data, providing an opportunity for reflexibility and credibility in interpreting results. Credibility was further established through the use of extensive quotes from participants to support our presentation of the findings. We established transferability by providing a thorough description of the sample and the context, and provide a detailed description of our methodology and analysis to ensure dependability.

*2.4. Ethical Considerations*

The Mozambican *Conselho Nacional de Bioética em Saúde* (National Health Bioethics Committee) provided ethical approval for this study. Participation was voluntary, and all participants provided written informed consent. In the case of minors (under 18 years of age), participants provided their assent for participation, and consent was obtained from a parent, guardian, or other trusted adult. To ensure confidentiality, participants were identified only by codes, and at no time were their names recorded. Any data that could connect the participant to the interview, or enable identification of other community members, were erased from transcripts. All data were stored in password-protected folders accessible only to ICRH-M and Ipas team members involved in data analysis.

## 3. Results

*3.1. Participant Characteristics*

In total, we conducted 19 FGDs with adult community members, 19 triad interviews with adolescent girls and young women, and 15 individual IDIs with young women with induced abortion experience, with a total of 262 study participants. Details can be found in Table 2.

**Table 2.** Overview of study participants.

| Study Component | Province | Number of Discus-sions/Interviews Conducted | Total Number of Participants | Description |
|---|---|---|---|---|
| Triad interviews with young women and girls | Nampula | 10 | 30 | Girls from 15 to 23 years of age; students; single and married; with and without children. |
| | Zambézia | 9 | 27 | Girls from 15 to 20 years of age; students; single and married; with and without children. |
| Focus group discussions with adult community members | Nampula | 10 | 100 | 7 groups of women, between 22 and 52 years of age; single, married, and widows; literate and illiterate; with children; housewives, midwives, businesswomen, traditional medicine practitioners. 3 groups of men between 24 and 64 years of age; single and married; with and without children; students, informal and formal sector workers, unemployed. |
| | Zambézia | 9 | 90 | 6 groups of women from 18 to 70 years of age; married and widows; with and without children; housewives, midwives, students and out-of-school young women. 3 groups of men between 19 and 88 years of age; single and married; with and without children; students, informal sector workers, unemployed. |
| In-depth interviews with young women with abortion experience | Nampula | 9 | 9 | Young women aged 18 to 24 years; single and married; literate and illiterate; with and without children; students and out-of-school. Two had the abortion out of the health facility (HF), and five initiated the abortion in the community and then went to the HF. |
| | Zambézia | 6 | 6 | Young women and girls aged 15 to 21 years; single and married; with and without children; students. One had the abortion in the HF, three out of the HF, and two initiated the abortion in the community and then went to the HF. |

*3.2. Key Themes*

Below we present four themes and various subthemes that emerged from the data in relation to how gender values, norms, and practices affect young women's and girls' experiences of pregnancy and abortion, including their care-seeking behavior.

3.2.1. Theme 1: Gender Norms and Expectations Put Girls at Increased Risk of Unintended Pregnancy

In the context of discussions about girls' and young women's motives for seeking abortions, study participants shared their thoughts and experiences relating to the circumstances in which unintended pregnancies occur, revealing many contributing gender-related factors.

High Social Value of Motherhood

Underlying many harmful gender norms and practices is the high social value placed on fecundity and motherhood in society. The following quotes illustrate the expectation that girls should aspire for motherhood and marriage, and that sex should take place within this context, as well as the foundation of this value in religion:

*"[The neighbours would say] She [Fatima, the fictional character] doesn't get married, just plays around, she is a prostitute. She only studies, is always inside … with a computer, her life is only about papers. She is refusing, showing off, instead of finding a husband, having children … she should look for a marriage."* (Triad interview participant, Nacala)

*"God created you to give birth."* (Triad interview participant, Nacala)

This underlying value means that girls often desire pregnancy, either driven by social pressure to procreate or due to their own hopes and expectations around marriage and commitment from their partner, which they anticipate will be consequences of the pregnancy. However, despite initially wanting to become pregnant or welcoming the pregnancy when it happens, it was reported that many girls find themselves abandoned by the boy or man who impregnated them if he refuses to assume responsibility for the pregnancy, which can lead to the girl regretting the pregnancy.

*"She is afraid. Who will buy the clothes? Who will support the child and provide food? So, if you regret being pregnant just take it out. The father made me pregnant and now he is denying it."* (Male FGD participant, Mugeba)

Factors Preventing Contraceptive Use

Adolescent girls who are not planning to become pregnant may not be able to effectively use contraception for a range of reasons, many of which are tied to gender roles and expectations, which constrain their reproductive autonomy.

Girls revealed gaps in knowledge about the circumstances in which pregnancy occurs and how to effectively prevent it. They demonstrated misinformation and a lack of accurate knowledge about contraceptives, how to use them and how to access them, and a reliance on methods that are of low effectiveness, such as coitus interruptus, washing the vagina after sex, or using contraceptive pills only on the day they have sexual intercourse.

*"Me, for example, when I know that I'm going to meet my boyfriend, I take pills that same day, or when I go without taking those pills on the day I come back, I take a pill the same day."* (15-year-old girl who had an abortion, Quelimane)

Girls also reported a lack of autonomy to take decisions about using contraception. Parents often do not allow their daughters to use contraception, with their opposition rooted in the social value of motherhood, for example, if they think contraception can cause infertility or because they believe it is women's role to bear children. As well as parental opposition, girls report that many male partners refuse to wear condoms during sex, or say they will withdraw before ejaculation and then do not. Asked why girls might not use contraception even when they have access to information, a participant replied:

*" … if the boyfriend doesn't have [a condom] or she was prepared, she should say no … But sometimes they're shy and let themselves be taken, they [the boys] say 'if you don't give it to me today we're finished', and so they're tricked."* (Triad interview participant, Mogovolas)

Harmful Gender Practices Leading to Pregnancy

Early marriage is prevalent in the study sites, where social norms determine that girls are ready for marriage after menarche, at which point they are expected to prepare for getting married and having children, instead of pursuing their studies. Once married, girls are expected to rapidly become pregnant, regardless of their age. Gender-based violence is also common, including rape and abuses of power, such as sexual relations involving teachers or older men with economic power, and was mentioned as a cause of pregnancy in young women and girls.

*"Maybe for her to get pregnant she suffered a rape . . . was violated without her wanting that man, and from that she gets pregnant."* (Triad interview participant, Nampula City)

A lack of access to financial resources was reported to contribute to adolescent pregnancies, especially in the case of girls from poorer families who may enter into transactional sex in exchange for money or material goods, either on their own initiative or at the request or obligation of their parents to bring in money for the family. In many such cases, the resulting pregnancy is not assumed by the man.

*"Because the parents sometimes, with a 16-year old daughter, they force her to date even though she doesn't want to . . . . Lots of parents do this. So she'll start dating, to get soap, to get food . . . And when she's with the man then comes the pregnancy."* (Triad interview participant, Mogovolas)

3.2.2. Theme 2: Gender Norms and Stigma around Adolescent Pregnancy Affect Pregnancy Decisions

Participants felt or demonstrated high levels of stigma around pregnancy in adolescents and unmarried young women, which impact girls' decisions about whether to keep or terminate the pregnancy.

Adolescent Pregnancy Is Highly Stigmatized

Pregnancy in adolescence is very stigmatized and viewed as a negative event with severe consequences, particularly if it occurs outside of marriage. Pregnant girls face discrimination, insults, humiliation, and ostracization. They can be separated from their friendship group, expelled from their parents' home, and abandoned by their boyfriends. Girls who become pregnant may also be at an elevated risk of violence within their households, including physical abuse from parents or other family members as punishment. This stigma can lead pregnant girls to become even more economically reliant on their male partner, as their family and community turn them away.

*"There are other people who insult her, they say 'we told you to take care of yourself and now you see how you are; can you see it? You are dirty, you're covered in baby poo.' Then, they start insulting her."* (Triad interview participant, Mogovolas)

*" . . . there are mothers who forbid their daughter to play with her friend who is pregnant, they even say: 'I heard that your friend is pregnant, and you are playing with her. Tomorrow it will be you too, I don't want you to play with her'."* (Female FGD participant, Quelimane)

*"Some are expelled from home to go and live with the father of the baby; if the father doesn't accept it, she will live in the streets, under the trees..."* (Female FGD participant, Quelimane)

As a result, experiencing a pregnancy brings a tremendous amount of stress and anxiety to girls as they deliberate whether to continue with the pregnancy or to have an abortion, as well as how to have an abortion, and the social, emotional, and practical implications of their decision. This stressful situation can lead the girl to despair and even suicide.

*"It was very sad for me to make that decision because, firstly, I would be taking a life that was inside me; secondly, it is an act seen as bad in society, society still condemns*

*abortion, so it was very difficult for me because I was also afraid of having a haemorrhage and dying. It was not easy for me to make that abortion decision.*" (Girl who had an abortion, Nacala).

"*The girl ends up overwhelmed with so much suffering due to humiliation from her own family . . . so then she either ends up looking for one of those "godfathers" who will mislead her, or else she ends up committing suicide at some point, it happens, some commit suicide because they were pregnant and had no one to take care of her baby.*" (Male FGD participant, Quelimane)

Factors That Lead Girls to Decide to Continue with an Unintended Pregnancy

Young women and girls may decide to keep a pregnancy, even if unintended and maybe unwanted. Abortion stigma is high, and perceived negative views of abortion can dissuade girls from seeking one. Abortion stigma is linked to the belief that having children is an inherent part of being a woman, and therefore, girls must not have an abortion even when the pregnancy is not wanted. Another factor is fear of the consequences of abortion, in particular, the possibility of dying or becoming infertile, since it is commonly believed that abortion causes infertility, especially when performed outside of the health facility. The fear of infertility is linked to the high value placed on motherhood, which can lead others to influence girls to avoid abortion, as well as the belief that if a girl terminates her first pregnancy, she will jeopardize her chances of becoming pregnant again.

"*They say that children are our riches, so it's better to give birth than to abort, even when the pregnancy is unwanted, we have to have the baby.*" (Triad interview participant, Nacala)

Girls reported lacking information about their right to abortion and the availability of safe and legal options, and demonstrated a lack of knowledge about safe abortion services and how to access them, which could influence their decision making about unintended pregnancy.

In the case of married women and girls, it is generally believed that they should not terminate their pregnancy, as they have a partner to support them and should bear children following the example of their own mothers; they may therefore be under additional pressure to keep the pregnancy.

"*She must not do it because she is already married, the baby will have a father that will provide for her, because the husband exists.*" (Triad interview participant, Mocuba)

Financial aspects also come into play in decision making. There are situations where the girl initially decides to have an abortion but is unable to go through with it because she cannot access the money needed to do so. Furthermore, women's and girls' low economic empowerment means that they are often financially dependent on a male partner, adding to expectations that young women should find a husband to support them, and pregnancy can be seen as a way of getting or keeping a husband. The longer-term economic benefits of having a child are also used as an argument to encourage girls to keep a pregnancy, as children are seen as a future resource that will support their parents in later life.

"*In the same way that I raised and gave birth to her, she must also do the same because today or tomorrow I can die and she might be alone, so, who will take care of her when she becomes ill? She must have and raise children.*" (Female FGD participant, Mocuba)

"*... She gets married to her husband, that is a reward for a poor woman . . . so she got married, legalized, she cannot take out pregnancy because it is a reward.*" (Male FGD participant, Nampula City)

Despite the stigma around unintended adolescent pregnancy and perceived lack of support from family, partners, and friends, it was seen that, in practice, some social support is available to girls, and this can influence them to decide to continue with the pregnancy. Girls in this situation are often economically supported by their parents, especially after the baby is born, and some return to school while the baby is raised by relatives, usually

their mother. When the boyfriend assumes responsibility for the pregnancy and the child, she may decide to keep the pregnancy even without it being planned; in these cases, he may also help with the expenses. However, even with this support, it is still common for girls to face negative impacts, such as dropping out of school to earn income to help the family with expenses.

> *"It's the parents that will be caring for her"* . . . *"the parents help her because she won't manage by herself."* (Triad interview participants, Mogovolas)

Factors That Lead Girls to Decide to Have an Abortion

Despite the significant stigma and perceived risks associated with abortion, for many girls the stigma of unintended pregnancy was so great that abortion was considered a less stigmatized and socially safer option.

> *"It's me who decided to do that, I saw that my child would be small, and I also was young, and I felt ashamed in the neighbourhood . . . and the neighbours . . . I was afraid they would laugh at me... saying look at that child, she already got pregnant . . . . that's why I got rid of my belly, I went to look for my friend and she accompanied me, and they gave me this medicine and I took it."* (Girl who had an abortion, Nacala)

Besides the stigma, pregnancy can have serious practical consequences for girls, fear of which leads them to choose to terminate their pregnancies. Parents can force pregnant daughters to marry the person responsible for the pregnancy, even if they do not want a marital relationship with him (for example, because they are still young or studying). Pregnancy frequently leads to girls dropping out of school, which can result from being prohibited from studying by their partner or parents, needing to work to support the child, or childcare demands. Pregnant girls may also decide to drop out due to feeling unwelcome at school, with discrimination by teachers and schoolmates reported as common, as well as girls being switched to night shifts by the decision of the school board. For girls who are in school, fear of the impact of pregnancy on their studies was one of the main reasons cited for seeking an abortion, as they believe that very few pregnant girls are able to continue to study.

> *"I did not even want it, I wanted to study, but I'm pregnant and if my mother takes me to the man's house I'm not going to study anymore.... Because they take you to your husband's house, the one that made you pregnant... and he accepts her, it's compulsory."* (Triad interview participant, Mogovolas)

> *"What motivated me more, more and more was thinking about my studies, I began to think that I would stay for many years without studying, my friends would be ahead of me, and I would be behind them."* (Girl who had an abortion, Quelimane)

Linked to the stigma of pregnancy and fear of the consequences, another very significant factor is the fear that girls have of their parents' reaction if they find out that they are pregnant, which can be violent and severe. Dialogue with parents about sexuality and SRHR issues is weak, and it is common for parents not to know when their daughter has a boyfriend and becomes sexually active. Some girls mentioned that they would not want to disappoint or let down their parents with an early or unplanned pregnancy. This dynamic with parents results in girls preferring to hide the pregnancy and have an abortion so that they do not have to face their family.

> *"I was afraid because in my house they would tell me that it happened because I was not listening to what they told me... so, to avoid mockery I took it [the pregnancy] out."* (Girl who had an abortion, Nacala)

> *"I would take it out [have an abortion] because if I go home to talk about this thing here, my parents, my parents . . . will kick me out of the home."* (Triad interview participant, Nacala).

It was reported as common for men and boys not to accept responsibility for pregnancies, abandoning girls without financial and other support. In these situations, abortion was said to be a common decision, and generally supported by others, particularly in the case of the man or boy being an "official" boyfriend, or when it is perceived that the girl has been deceived by a married man.

*"... [when the girl becomes pregnant] the boys refuse to assume responsibility for the pregnancy, sometimes they even flee from their city or community, then the girl gets overwhelmed, she becomes unprotected and feels humiliated."* (Male FGD participant, Quelimane)

*"I was studying.... I started to play and I got pregnant, I didn't know that he was married... He was cheating me saying he wasn't married, but later... I discovered that he was married. I wanted to take the belly out because he didn't want to take responsibility for it, then I tried to cause an abortion."* (Girl who had an abortion, Nampula City)

For women and girls in a relationship, if the pregnancy occurs in a context of marital problems, particularly intimate partner violence, this can lead a woman to choose to abort rather than go through with a pregnancy. Examples were given of cases of physical violence, economic violence (lack of financial support), and psychological/emotional violence (including infidelity).

*"I decided to abort because I was not doing well in my house with my husband, he also used to come home late, so I thought that raising that child would not be a good thing... We were not getting along at home, life was just arguing, fighting, so I thought I'd better have an abortion."* (Young woman who had an abortion, Mogovolas)

*"...lack of understanding at home, for example, if I'm at home with my husband but we are in constant arguments, then, the thing [pregnancy] happens and I discover it alone, my husband who lives with me doesn't know, I will think about aborting because we are always fighting, almost every day he beats me, so, before he kicks me out of the house and I suffer with my pregnancy, I prefer to have an abortion."* (Female FGD participant, Quelimane).

Abortion is also seen as a preferred (and more socially acceptable) option by some if a pregnancy is a result of rape.

*"When the person is raped, they should not have that child.... Because when the baby is born, the more the mother sees the child, she remembers that horrible thing that happened to her, so she won't be able to be happy. Often due to that they want to have an abortion."* (Triad interview participant, Nacala)

Finally, girls may choose to have an abortion if they feel they are too young and not ready to have a child. While girls are often considered to be ready to marry and become mothers once they have started menstruation, others believe that minors are not old enough to have a baby as they are still children themselves. Girls do not feel prepared to be mothers because it brings responsibilities that they feel unable to cope with and means that they will have to grow up fast, or because they do not feel ready for the new role they would have to play in society as a mother and the resulting changes in their lives, such as feeling less attractive and having to stop playing and dating.

*"She doesn't even want to be disturbed being called 'Mum, Mum', she doesn't want it... she doesn't want to hear someone calling her 'mother of [...], mother of Maria', she just wants to be called by her own name."* (Triad interview participant, Mogovolas)

*"I would think, I'm pregnant, my friend is not, it's better to have an abortion, for us to be on the same level and enjoy life... Me being old and my friend young? No way, I wouldn't accept it and that's what would make me think about having an abortion."* (Triad interview, Mugeba)

### 3.2.3. Theme 3: Gender Norms Contribute to Abortion Stigma and Influence Choice of Provider

Abortion Stigma and the Influence of Gender Norms

Abortion is seen in the community as something unwanted and sinful, which runs against traditional gender roles; specifically, it is contrary to community expectations of women's and girls' roles as child bearers. Girls who have had an abortion are seen as having lower value as a woman.

*"She is not a woman because she killed and got rid of her belly. If a man goes to her, like he wants to go out with her, others will say 'that one wants to kill your baby'."* (Triad interview participant, Mocuba)

*"[In the community] they would think and speak ill of her because she committed that act of abortion which is not accepted in the community, and they would also think that what she did is very bad, without thinking why she did it, they would judge without even knowing her."* (Triad interview participant, Nacala)

Abortion is also associated with promiscuity and women's irresponsible behavior, which is seen as another manifestation of women behaving outside of gender expectations. Community members consider that most abortions take place in young and single women and assume that this is because they are irresponsible and "shallow", also often assuming that girls who have abortions have multiple partners and do not know who made them pregnant.

*"These unmarried women do this [abortion] because they don't control themselves, because if you have only one boyfriend, when you get pregnant you already know who the father is, and he cannot run away and refuse the pregnancy."* (Female FGD participant, Mocuba)

*"She managed to conceive, but there was no one to assume responsibility for it or take care of it, she ended up aborting, so she is a prostitute, she ends up being a prostitute. . . . Who does it [the pregnancy] belong to? If she . . . stayed up from late night to the end of the dawn, she went out . . . then, she loses value of being considered a good, polite girl who is not a prostitute."* (Male FGD participant, Quelimane)

On the other hand, more socially acceptable or understood reasons for abortion include if the partner does not assume responsibility or abandons the pregnant woman, if the girl has been "tricked" by a married man, if the pregnancy is a result of rape, if the girl does not have the economic means to raise a child, if she does not want to interrupt her studies, or if it is her first abortion. There was some evidence that in these cases, women and girls faced lower levels of abortion stigma.

*"We can say that she was right, to take it out [have an abortion], because she didn't have a secure place or her boyfriend denied it."* (Triad interview participant, Quelimane)

*"Cases of rape, in these cases the community can say that she is right [to have an abortion], she couldn't accept having a child that the father doesn't know."* (Triad interview participant, Nacala)

Experiences of Abortion Stigma

Girls who are known to have had abortions are humiliated, marginalized, and stigmatized by their community. Parents' reactions to discovering that their daughter had an abortion can be severe, including throwing the girl out of the home and violence. Other girls' parents may see girls who have had abortions as a bad influence and prohibit their daughters from staying friends with them. Many pejorative terms are used to describe women and girls who have had an abortion, including 'prostitute', 'disobedient', 'a shame on the family', 'dirty', 'murderer', 'has a cemetery in her stomach', 'stupid', and 'witch'.

*"Prostitute, she doesn't deserve to have another child, she didn't want the one God gave her . . . she didn't want to raise it and took it out, she doesn't deserve to have another child,*

*she deserves to stay like that forever . . . she's a slut."* (Triad interview participant, Nampula City)

*"Others would hate her . . . mainly her friends would push her away because of the influence of their mothers . . . a mother would say 'ahm . . . your friend's behaviour, I do not like it, you have to stop hanging out with her because you can be influenced to do what she did' . . . her friends would stop hanging out with her and that friendship would no longer be that strong because she did something very bad."* (Female FGD participant, Nampula City)

Stigma is also present at health facilities, with reports that some providers insult girls seeking abortion care; this stigma can be associated with abortion or with the fact that they are sexually active, unmarried adolescents. Girls mentioned that the provider may also try to persuade them to keep the pregnancy (although this was not common). Although MoH guidelines do not require women who seek an abortion to involve their partner, in practice, some providers require the partner to be present or request consent from the partner or parent.

*"Hmmm, they charge, and she doesn't have money. Or those nurses can insult her, 'When you got pregnant didn't you know? When you had sex without protection didn't you know that the consequence would be pregnancy? . . . Why do you want an abortion, doesn't this child have a father, did you do it by yourself?' And because she wants to run away from this, from all these steps, she can think it's safer to do it outside [of the HF]."* (Triad interview participant, Nampula City)

Impact of Gender Norms and Other Factors on Choice of Provider

Gender-related factors influence women's and girls' choice of abortion provider, place, and method and, in many cases, lead them to seek abortion care from providers and methods outside the formal health care system. Given that girls often want to conceal their pregnancy and/or abortion from their family and others, the most cited concern was the need for privacy and confidentiality. In general, girls felt that abortion care outside the health facility was preferable in this regard. Many of the participants' abortion experiences involved initially attempting to induce an abortion themselves at home using conventional medication, plants, or common household products, such as cola, salt, and detergent. Others involved an informal provider, usually either a health worker at their home using conventional abortifacient medicine or a traditional medicine practitioner using plant-based treatments or inserting a sharp object, such as a cassava stem, into the vagina. It was generally felt that health workers did not respect confidentiality, and that girls would be seen by others at the health facility. In this context, a "safe abortion" is one that reduces the social risk associated with abortion, where the girl feels well treated and cared for in a private and confidential environment.

*"[Teenagers] go to the healer because normally healers are neighbours, right? They prefer the traditional healer because if they go there with a fifty [50 Meticais] and say auntie, please help me . . . and the healer will give help right away, but in the hospital... [laughs] first they will tell you to bring your husband, then they will insult you right there... [while healers] do not ask you many questions because their intention is to get money, so they never ask where your husband is, etcetera . . . "* (Triad interview participant, Mugeba)

*"They usually think like . . . let me take medication for people not to find me, or if I go to the hospital, someone will see me. Now, for someone not to find me out, I want to go to the market or the pharmacy and buy the pills, do the things I know, and people will only find out after I have done it."* (Triad interview participant, Mocuba)

Young women's and girls' lack of financial resources affects their choice of provider. The perceived cost of safe abortion (in the health facility) is much higher than the other

alternatives. Abortion outside the health facility is always considered more affordable and can often be paid in kind.

*"I asked about it and they told me that if you go to the hospital you'll spend a lot, so it's better to look for a traditional healer."* (Girl who had an abortion, Mogovolas)

Women's and girls' lack of access to accurate, up-to-date information about safe abortion options, their right to access services, and how to access them may be contributing to their nonuse. One group of girls referred to "those prohibited pills" when talking about abortion at the health facilities, and others mentioned that abortion is a crime and that a girl who has one is at risk of being arrested.

*"I did it on my own at home because I was afraid that in the hospital they would refuse doing it for me, so I decided to take the risks alone."* (Girl who had an abortion, Nampula City)

3.2.4. Theme 4: Men, Family Members, and Others Have a Strong Influence on Pregnancy and Abortion Decision Making

Some participants consider that decisions about terminating or maintaining pregnancy can be made by the girls themselves, and some girls who have had abortions testify that they made the decision on their own. However, even when the decision is apparently made by the girl, it was seen that in fact she takes it based on the social context in which she lives, which sometimes leaves her with no other option, and is based on the opinions and advice of people around her. Key influencers include mothers, fathers, uncles, aunts, friends, sisters, and boyfriends—people who usually have a degree of power over girls by virtue of their age, gender, and/or social standing; they can either make the decision directly, or they pressure the girl into deciding in a certain way.

Involvement of Others in Decisions about Pregnancy Outcome

Male partners often play a role, particularly if they do not want or feel ready to take responsibility for the pregnancy, or if they are already married. They may take the abortion decision directly, or may effectively force the girl to decide to abort by refusing to accept responsibility for the baby.

*"Well, after I looked for my partner and I told him that I was pregnant, he said that we had to terminate the pregnancy because he didn't have the means because he was still studying . . . . and because I was also studying, and my parents didn't know that I was pregnant. Also, I couldn't have the baby alone, when my partner had already refused."* (Girl who had an abortion, Mocuba)

*"Sometimes they [the men] say if you don't want to abort, you will bear the consequences alone."* (Triad interview participants, Quelimane)

*"It's my boyfriend that decided. He said, 'we'll take it out because we're kids, we can't have children this early'. I accepted, he took out money, and came with me to the hospital."* (Girl who had an abortion, Mocuba)

The girl and her boyfriend can also make the decision together. Girls often talk first to their boyfriend about the pregnancy, and they may decide together whether they will involve their parents or whether they will resolve the matter and make the decision on their own. The girl may need to involve the boyfriend in order to get money to have the abortion without telling the parents. When male partners choose not to be involved in the decision-making process, this can contribute to further stigma for the pregnant girl, who may experience judgment from the community, including accusations that she does not know the father.

*"When I missed a month, I talked to my boyfriend, I said 'I'm pregnant'. He said 'you're pregnant, what are we going to do?' and I said, 'ah, I don't want to take it out.' He said 'let's take it out because I cannot help you, I'm studying' and then I said 'okay... I'm*

*studying too and my parents will not like it, I want to study first'.*" (Girl who had an abortion, Mocuba)

Many girls have the perception that it is necessary to consult the man responsible for the pregnancy before making the abortion decision, particularly if they are married. Some girls believe that for many married women or girls, abortion is not an option because they would need to have their husband's permission and he would not accept to abort.

There are cases where the parents or other family members make the decision. Both the girl's and the boy's family can be involved; for example, some reported family meetings held with the girl's and the boy's families to decide what to do, while others reported that the mother-in-law may take the girl to have an abortion. Parents may decide on an abortion if they think that the girl should continue her studies, or if the girl is very young.

*"It was my parents [who decided], they didn't want me to have that belly because I was young."* (Girl who had an abortion, Nampula City)

Involvement of Others in Abortion Decision Making

As with the decision about whether to terminate the pregnancy, male partners and family members are often influential in decision making about the choice of abortion provider and method. While women generally have some information about formal and informal services, providers, and methods available, men usually are much less well informed, with low levels of knowledge or incorrect information.

*"A 13-year-old girl became pregnant, and the boyfriend lived with his mother and did not want his mother to know that he had impregnated a girl, and the pregnancy was already 3 or 4 months, so he bought what he bought, alone, and gave it to the girl."* (Triad interview participant, Nampula City)

*"Well, I asked my mother what I should do, and she said we should go to the hospital, and then there at the hospital they will tell us what we have to do."* (Girl who had an abortion, Mocuba)

## 4. Discussion

We found that many gender-related barriers operate at different levels to negatively affect young women's and girls' experiences of pregnancy and abortion. Gender norms shape and constrain girls' access to abortion care, and intersect with stigmas around sexuality, adolescence, contraception, and abortion to harm girls' health and well-being. We conclude that that if these gender-related factors are not addressed, they will continue to drive many to seek unsafe abortions, even in a setting with partial decriminalization of abortion and an expansion of safe abortion service provision.

*4.1. Key Findings and Comparison with the Literature*

Gender-related issues from across the dimensions used for the analysis were identified, particularly in the dimensions of assets and resources, cultural norms and beliefs, and power and decision making, and are compared below with other findings from the literature, particularly from sub-Saharan Africa.

*Cultural gender norms and beliefs:* The high social value of motherhood was an underlying driver of anticontraception and antiabortion views, as well as encouraging girls to aspire to become pregnant and get married, even at an early age. Other authors have found that married women with children have a higher social value than other women and that aspirations for the status associated with motherhood and marriage can result in nonuse of contraception [47–49]. Others have noted that norms around motherhood determine which kind of pregnancy is considered unwanted, with those that go against traditional notions of motherhood and family often ending in abortion [50]. A common example of this seen in our study is girls who become pregnant outside of marriage, particularly when the man responsible for the pregnancy is not known or does not assume responsibility,

contravening the broader social norm that dictates that children should be raised in a family, with a father.

Abortion stigma was found to be high, sustained mainly by social and cultural norms and expectations, including those related to gender, and drives girls to seek unsafe methods and providers that they consider to be socially safe, including self-induced abortion. Similar findings have emerged from many other settings, which have found younger unmarried women to be particularly susceptible to abortion stigma, often linked to gendered attitudes and beliefs, such as the importance of procreation or the preconceived idea that girls seeking abortions have multiple sexual partners [17,30,51].

Stigma around premarital adolescent pregnancy was also very high, and consequently, girls who become pregnant find themselves facing a very difficult choice with little support, often leading them to make poor decisions. Stigma associated with adolescent sexual activity (particularly outside of marriage) and its consequences is common across the region [9,10,48,52–54]. High levels of worry and distress in pregnant adolescents have been seen elsewhere, principally due to the social and practical consequences of their situation [55–57]. Some authors have concluded that abortion is considered to be less shameful and to have lesser negative consequences than unintended pregnancy [58,59], and many have concluded that the resulting need for secrecy is a key driver of unsafe abortion [10,60,61], with "social safety trumping medical safety" [62]. Stigmatizing attitudes towards adolescent sexuality and pregnancy are often not addressed by adolescent SRHR programming, which tends to focus on the prevention of pregnancy; if this scenario continues, it is likely that girls will continue to resort to unsafe abortions that they consider to be more discrete.

Gendered values and norms underlie harmful practices, such as intimate partner violence, sexual violence, transactional sex, and early marriage, that were seen to increase young women's and girls' risk of unintended pregnancy and can lead them to opt for abortion. These harmful gender practices have been found in other settings to be a common motive for seeking an abortion [52,63,64], and violence has also been reported as a consequence of abortion [65]. Gender-based violence including early marriage is common in Mozambique, and we found that girls' fear of being forced to marry the man who impregnated them was a strong motive for them to seek an unsafe abortion in secrecy. Although pregnancy as a precursor for early marriage has been reported in other settings [66,67], we were unable to identify other studies that identified fear of forced marriage as a factor influencing the decision to abort, which merits further investigation.

*Access to assets and resources:* We found that economic issues often increase adolescent girls' risk of unintended pregnancy, a finding echoed by other studies that have reported transactional sex to be a driver of early onset of sexual activity [68], and women's and girls' financial dependence on men limiting their ability to negotiate contraception [59]. Many studies have identified socioeconomic reasons as some of the main motives for abortion, particularly among younger and unmarried women [61,68,69]. Young women's and girls' lack of financial resources has been reported in other settings to limit their choice of abortion method and provider [31,60,70]. Although safe abortion services are officially free of charge in Mozambique, we found that the perceived cost was high, and experience from Zambia shows that informal charges can be common as health providers exploit low levels of information and the desire for secrecy and discretion [60].

We found that access to information was important, with a lack of accurate knowledge about pregnancy and contraception contributing to the risk of unintended pregnancy and a lack of information about abortion rights, the legality of abortion, and how to access safe abortion services contributing to young women and girls choosing informal providers over formal ones. Another study in Mozambique came to a similar conclusion [30], and other studies have also concluded that younger unmarried women are particularly affected by a lack of information that leads them to make poor SRH choices and to have reduced access to services [51,60,71,72]. Limited knowledge about pregnancy and abortion, in combination with other factors, such as the need for financial resources to pay for services and waiting

for the partner to take a decision, can lead to adolescents delaying abortion [59,60,73]. These factors were observed in our study, meaning that it is likely that, in Mozambique, many adolescents and younger women would be ineligible for a legal abortion by the time they seek one.

Women's and girls' social assets were also found to be key, with support (or lack of) from family and partners heavily influencing pregnancy and abortion experiences. A lack of a stable relationship and partner support for a pregnancy has been identified by others as a common factor leading to the decision to have an abortion [59,61,68], and several studies have found that support from partners, family members, peers, and others can increase access to safe abortion services [60,70,74].

*Power and decision making:* We found that most young women and girls have little decision-making autonomy, with male partners' and family members' involvement in girls' decision making about contraception, pregnancy, and abortion often significant, and sometimes mandatory. Many other studies have reached a similar conclusion, finding that family members, partners, and providers exert direct or indirect influence over young women's and girls' decisions, sometimes using coercion or force to do so [61,63,74,75]. Women's and girls' lack of decision-making power was often linked to their lack of financial autonomy through their economic dependence on partners or parents or other power imbalances [30,59,68]. Despite men's key role, and evidence that they have high levels of misinformation and abortion stigma [76,77], few abortion programs significantly engage men and boys [78].

On the other hand, the decision to have an abortion has been identified as an expression of agency for young women, who have in some settings been able to decide to access safe abortion services as a way of retaining the possibility of more autonomy in the future (for example, through continuing education or work), despite the abortion stigma they face, with access to information and peer support identified as important in enabling this [63,70].

*Legal change* vs. *social change:* Based on our findings, we conclude that despite abortion law in Mozambique being significantly liberalized, social change that allows women and girls to access safe abortion services is yet to occur. It is generally accepted that although legal reform on abortion is an essential precursor to safe abortion access and can be a driver of social change, a number of other factors influence the extent to which this happens [14]. Legal reform is not always accompanied by timely implementation and expansion of quality services [79], and failure to inform women, health providers, and others about the legal change means that levels of knowledge about legislation and services are often low [80,81]. Persisting abortion stigma can negatively affect service provision and access [35,82]. Furthermore, our study showed that gender norms and the resulting stigma around unintended pregnancy are important drivers of unsafe abortion, and there is evidence that this type of social norm may be particularly resistant to change. For example, in Ethiopia, where abortion law was liberalized in 2005, a recent study found that although attitudes towards abortion have softened, unintended and premarital pregnancies remain highly stigmatized, continuing to drive women and girls to seek unsafe abortions [57].

*Likely applicability in other settings:* The research cited above demonstrates that strong gender norms such as those identified in this study in northern Mozambique are prevalent across sub-Saharan Africa, and similar themes have been found in other regions, particularly in low- and middle-income countries (LMICs) but also in higher-income countries [31,64]. While Mozambique has one of the most liberalized abortion laws in Africa, our findings demonstrate that legality is insufficient for guaranteeing abortion access, and that gender-related drivers of unsafe abortion are likely to be similar regardless of the legal situation and the existence of safe abortion services.

### 4.2. Recommendations and Future Directions

The findings from this study add to the body of evidence that indicates the need for SRHR programming to integrate multisectoral, gender-transformative approaches that

work at different levels to shift the harmful gender norms affecting women's and girls' experiences and improve their access to information and services.

Based on the findings of this study, adolescent girls need to be empowered with knowledge and self-efficacy on their SRH rights, particularly in relation to contraception, GBV, early marriage, and safe abortion care, as well as available services and how to access them. Girls who become pregnant need to be supported through pregnancy and helped to negotiate the challenges faced in this period and the options available to them.

Family members and male partners are key influencers, and strategies to engage men, boys, and parents need to be expanded and integrated into safe abortion and adolescent SRHR programming. This should involve promoting nonjudgmental communication between parents and children on issues relating to sexuality. Work with boys should include discussion of rights and responsibilities related to pregnancy, parenthood, and abortion.

Whole-community approaches—such as community mobilization activities, social and behavior change communication, and working to engage local leaders—are needed to reduce stigma around adolescent sexuality, premarital pregnancy, and abortion and challenge and shift harmful social and gender values, norms, and practices. Further work is needed with health providers to ensure that they are not reinforcing negative gender norms and stigma and understand the barriers young women and girls face in accessing contraception and abortion care. Involving young women and girls in monitoring SRH service implementation would help with identifying and addressing barriers and improving the quality of care.

At the programming and policy level, there is a need for adolescent SRHR programs to look beyond individual behavior and health information and more effectively address adolescents' social and cultural context, especially the gender values, norms, and practices that are so influential. The issues identified here are broad and would be best addressed through integrated, holistic approaches that bring together strategies to increase access to safe abortion, reduce early marriage, prevent GBV, promote gender equality, and provide adolescents with SRHR education and services. Strategies need to be long term, locally driven, context specific, and culturally appropriate, taking on board lessons learned in terms of what works to shift gender/social norms [41].

In terms of further research, it would be important to conduct research in Mozambique with adolescent boys and young men given their influential role in decision making. Research with healthcare providers to assess gender norms and stereotypes within SRH service provision would provide further insight into how this affects service quality and access. Implementation research is needed to build evidence on effective interventions that address sexuality, pregnancy, and abortion stigma and its impact on young people, including access to care.

### *4.3. Study Limitations*

This study took place in areas with strengthened CAC services through the partnership between Ipas and the MoH, and the results may not be generalizable to other contexts where safe abortion services have not yet been officially introduced, including populations in areas far from health facilities. Furthermore, since data were collected in 2018, there have been advances in the expansion and quality of CAC service provision and community awareness raising, although gender norms are unlikely to have shifted significantly during this period. The study findings may not be transferable to women older than those targeted by the study (up to 25 years old), although many findings regarding married young women are likely to be applicable to older married women; similarly, the perspectives of adolescent boys and younger adolescent girls were not covered by the current study. Transcripts were translated from local languages into Portuguese, and results are presented here in English; thus, despite our multilingual study team's efforts, it is possible that some nuance could be lost in translation. The original study objectives were not focused on gender norms, and were focused mainly on abortion and not pregnancy, making it possible that some issues were not picked up through the interviews.

## 5. Conclusions

There is still a lack of data on SRH experiences of adolescent girls, particularly around abortion, and available data often do not consider the role that gender inequality and norms play in affecting their experiences and access to care. To understand better how gender values, norms, and practices may be influencing access to care in a setting where safe and legal abortion services are available, we analyzed data from a formative qualitative study conducted in Mozambique.

Norms and stigma around abortion, adolescent sexuality, and unintended pregnancy interact with unequal power relations, limited decision-making autonomy, and poor access to and control over resources to strongly influence the occurrence and outcome of unintended pregnancies and abortion and inhibit young women's and girls' access to SRH services, including contraception and comprehensive abortion care. Consequently, despite the liberalization of legislation and the introduction of safe abortion services, many girls are likely to continue to use unsafe options in Mozambique and similar settings where these gender values, norms, and practices are strong. While there is increasing evidence of the important role that gender-related factors play in adolescent sexual and reproductive health, this is still often not effectively addressed in programming.

Gender-transformative approaches involving challenging gender norms, empowering girls to make informed decisions, strengthening partner and family support, and ensuring that providers do not impose barriers could increase access to safe abortion and other SRH services. Increasing access to safe abortion could also be key to increasing gender equality, as lack of access perpetuates inequality due to the substantial negative impacts of unintended pregnancy and unsafe abortion, which include early marriage, dropping out of school, and death and disability.

**Author Contributions:** Conceptualization, S.D. and S.G.; methodology, S.D., E.M., S.G. and M.d.M.; validation, N/A; formal analysis, S.G. and J.J.P.; data interpretation, all authors; investigation, M.d.M. and J.J.P.; resources, N/A; data curation, S.G., M.d.M. and J.J.P.; writing—original draft preparation, S.G. and G.S.; writing—review and editing, G.S., S.D., E.M. and J.M.; visualization, N/A; supervision, S.G., S.D. and E.M.; project administration, S.G. All authors have read and agreed to the published version of the manuscript.

**Funding:** This research was funded by Sida (Swedish International Development Cooperation Agency), Contribution ID: 51140118.

**Institutional Review Board Statement:** The study was conducted in accordance with the Declaration of Helsinki, and approved by the Conselho Nacional de Bioética em Saúde (CNBS, National Health Bioethics Committee) on 15 May 2018 (protocol code 157/CNBS/2018).

**Informed Consent Statement:** Informed consent was obtained from all subjects involved in the study.

**Data Availability Statement:** The research protocol and data presented in this study can be made available upon request from the corresponding author. Data are not publicly available due to confidentiality agreements with participants.

**Acknowledgments:** The research team wishes to thank the following individuals for their contributions: Karmen Assura and Alexandra Teixeira from Ipas for research conceptualization and design support; Emídio Cumbane at ICRH-M for data management; and Celestina António, Angelina Tomás, Maria Tomás, Cácia Manuel, and Leonarda Varinde for data collection. The team also acknowledges the important contribution of the Provincial Directorates of Health and Provincial Directorates of Youth and Sport of Nampula and Zambézia provinces; District Services of Health and Social Action and District Services of Youth and Sports of Rapale, Nampula City, Nacala Porto, Mogovolas, Quelimane, and Mocuba districts; Coalizão da Juventude Moçambicana in Nampula and Zambézia provinces; and the study participants who shared their knowledge and experience.

**Conflicts of Interest:** The authors declare no conflict of interest. The funders had no role in the design of the study; in the collection, analyses, or interpretation of data; in the writing of the manuscript; or in the decision to publish the results.

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
