# Peer review of "The Role of Gender Norms in Shaping Adolescent Girls’ and Young Women’s Experiences of Pregnancy and Abortion in Mozambique"

_adolescents, doi:10.3390/adolescents3020024_

Round 1
Reviewer 1 Report
Comments and Suggestions for Authors
Thank you for the opportunity to review the manuscript titled “ The Role of Gender Norms in Shaping Adolescent Girls’ and Young Women’s Experiences of Pregnancy and Abortion in Mozambique”
Overall, the work is well organized and comprehensively described, and is expected to have a significant contribution to our knowledge on youth health in understudied and underprivileged populations.
Here are some areas for improvement:
1- Contextual information is much needed. It is very important for readers and researchers to understand the context of this city and characteristics of its population. This can also help us gauge the generalizability of the findings. The current introduction goes prematurely into presenting abortion in Mozambique. What about the country itself? Homogeneity of its population, demographic characteristics, socioeconomics, health culture, differences across provinces?
2- The data analysis process should be clearly named and described. This should also be cited as there are many ways to analyze qualitative data.
3- I understand that two female researchers and four young (under-25) female research assistants. What’s the difference between the two groups and what tasks were conducted by each. Who trained the data collectors and how their work was monitored?
4- Were study methods amended as data collection and results were emerging?
5- Please add a clear section on how the rigor of the data was protected.
Thank you!
Comments on the Quality of English LanguageMinor editing of English language required. Especially the results section.
Reviewer 2 Report
Comments and Suggestions for Authors
This is a really interesting paper which considers access to abortion, approx. 10 years after it was partially decriminalised in Mozambique. The paper essentially argues that while there remain many barriers to safe abortion such as poor access to services, not enough girl-friendly services and lack of information, gendered social norms remain one of the most significant barriers – and it is not given the attention it deserves. The paper reports on an extensive qualitative study resulting in very rich data on the unwanted pregnancies and abortion. Overall, it is a really great paper. Well written and comprehensive with clear recognition of limitations.
I have a few small comments to make with the aim of improving the paper further, for publication.
I appreciate that qualitative analysis takes up many of the available words, however I feel like the specific context of Mozambique and its history gets a bit lost, and how this context and history impacts on the social norms held (i.e. how process of colonisation increases patriarchal values and power, and impact of long term conflict and economic fragility on gendered social norms). For example, you mention religion, a few times, but don’t explain the religious diversity across Mozambique. Also, while your participants may not have been affected by crisis at the time, Mozambique is at high risk of environmental and climate related crises, which also impacts how gendered social norms play out. A brief recognition of these factors and the implications for gendered social norms would strengthen your paper even if you don’t have any findings which speak directly to these issues.
On p3 l113 you state that ‘Few studies have examined the gender-related factors which are likely restricting access to safe abortion and other SRH services [36]. Furthermore, SRHR policy and programming for adolescents often does not take into account the impact of social and gender norms [4,37].’ I think this statement needs further unpacking (a little) as social norms are frequently mentioned as a factor impacting SRHRs and access to services and SRHR education in reports and best practice guidelines etc, which many readers may be familiar with – however despite it being a well-recognised issue, there are few studies that have done the in-depth work that you have done. So I think you need to expand on this a bit and explain why in depth studies are important even though the impact of gendered social norms is well recognised in the field of adolescent SRHR.
Other points that I think you need to engage better with are;
a) You state that 44% of 15-19 year-old girls become pregnant before turning 18, and 14% before they are 15. That’s over half of all girls. If adolescent pregnancy is so common (in and outside of marriage), why is there still such a stigma associated with it – if rates are that high then every other girl will be impacted by this stigma and shame and negative behaviour … and yet your participants don’t really convey this in the data???
b) We know that the relationship between legal change and social change is complex and non linear and debated, so it is not that surprising there has been little change since the partial decriminalisation of abortion, and it would strengthen your discussion to draw on the literature which explores the relationship between laws and social norms.
c) Your sample includes diversity in location, i.e. urban/peri-urban/rural etc, and I am unclear if your sample was diverse in other ways (i.e. by age or by religion or ethnicity or (dis)ability etc) and I wondered why you don’t make any references to diversity of experience or opinion by different identity markers.
d) Participants mention the need for the father to ‘take responsibility’ a lot – in a similar qualitative study that I also undertook on unintended pregnancy in Mozambique (about 15 years ago now), my participants referred to the father of the child as ‘o dono’ (the owner), and ‘the owner’ of the stomach during pregnancy. Your analysis of gendered norms predominantly focuses on the norms related to girls, and makes little reference to the norms related to boys and men’s when it comes to family formation and fatherhood and pregnancy, - these are intertwined of course, and I felt you could have said more about how broader social norms related to heterosexual family formation, and role of the father etc also play a big role and why the ‘claiming of responsibility’ is such a big part of the experience and why you can’t be a single mum and why every child must have a father.
I also think that you miss an opportunity in the recommendations and future directions section to get the best out of your research and to show why such in depth consideration of gendered social norms is critical for ensuring access to safe abortions. Many of the recommendations you make are already best practice and recommended practice for adolescent SRHR even if they are not implemented/operationalised (i.e. engaging with men and boys, engaging with communities, encouraging more supportive parental/child relationships, girl-friendly maternity and SRH services etc). You highlight the need for gender transformative approaches and also whole-community approaches in the recommendations, without defining these or explaining what you mean by these. I feel that you could draw on your rich findings to make much more detailed recommendations to enhance gender transformative approaches and also whole-community approaches.
Many of your recommendations narrowly focus on SRHR education and service delivery (which is obviously important) but we know that shifting ingrained gender norms requires holistic, significant, long term, multi-level, intentional, ‘locally’ driven work (top down and bottom up) – one sector can’t do this alone, and so I was surprised by your lack of recommendations to tackle the structural aspects of gendered social norms by working alongside other sectors, and also no mention of ‘African’, ‘indigenous’, ‘culturally appropriate’ interventions to change gendered social norms and values that prevent girls accessing safe abortions or prevent them from living empowered lives if they choose not to have an abortion.
Finally, because you have four main themes and then multiple sub themes – it would be helpful to include in the label theme 1, theme 2 etc when introducing each theme.
Round 2
Reviewer 1 Report
Comments and Suggestions for Authors
I thank the authors for revising their work and addressing reviewers' concerns and suggestions.
I believe the manuscript is much improved as a result of these edits and I have no further issues to raise.
Good luck!
Comments on the Quality of English LanguageNone
Author Response
Dear Reviewer,
We thank you once again for your comments that have certainly helped us to strengthen the paper.
With warm regards,
Sally Griffin